# The Pathways to Participation (P2P) Program: A Pilot Outcomes Study

**DOI:** 10.3390/ijerph19106088

**Published:** 2022-05-17

**Authors:** Danielle Hitch, Lindsay Vernon, Rachel Collins, Carolyn Dun, Sarah Palexas, Kate Lhuede

**Affiliations:** 1Occupational Science and Therapy, School of Health and Social Development, Deakin University, Geelong 3217, Australia; 2North Western Mental Health, Melbourne Health, Parkville 3052, Australia; lindsayjayne@yahoo.co.uk (L.V.); rachel.collins@austin.org.au (R.C.); carolyn.dun@mh.org.au (C.D.); sarah.palexas@gmail.com (S.P.)

**Keywords:** recovery, mental illness, mental health, psychiatry, social inclusion, occupational therapy, occupations, time use, activities of daily living, work

## Abstract

Research has consistently found that people with mental illness (known as consumers) experience lower levels of participation in meaningful activities, which can limit their opportunities for recovery support. The aim of this study was to describe the outcomes of participation in a group program designed to address all stages of activity participation, known as Pathways to Participation (P2P). A descriptive longitudinal design was utilized, collecting data at three time points. Outcomes were measured by the Camberwell Assessment of Need Short Appraisal (CANSAS), Recovery Assessment Scale—Domains and Stages (RAS-DS), Behavior and Symptom Identification Scale (BASIS-24), Living in the Community Questionnaire (LCQ), and time-use diaries. All data were analyzed using descriptive statistics and Chi-square analyses. A total of 17 consumers completed baseline data, 11 contributed post-program data, and 8 provided follow-up data. Most were female (63.64%) and had been living with mental illness for 11.50 (±7.74) years on average. Reductions in unmet needs and improvements in self-rated recovery scores were reported, but no changes were identified in either time use or psychosocial health. The findings indicate that the P2P program may enable consumers to achieve positive activity and participation outcomes as part of their personal recovery.

## 1. Introduction

The link between participation in meaningful activity and health or well-being has been well established through rigorous research studies across age and diagnostic groups [1]. For people with mental illness (known as consumers), participation has also been identified as a key facilitator for recovery [2] and full participation in community life [3].

However, a considerable body of evidence demonstrates that consumers participate in lower levels of activity than members of the general community. An Australian study of consumers living in the community [4] found they are mostly engaged in home-based activities and unpaid work, with consumers who were older, female, and from culturally and linguistically diverse backgrounds less likely to undertake some categories of activity. Less participation by consumers is a persistent issue, having been consistently identified in research throughout the past 20 years [5,6]. Multiple studies have found that consumers have a different pattern of time use than the general population, spending significantly more time sleeping, eating, performing self-care tasks, and doing quiet activities [7,8]. 

Decreased participation in meaningful activity has multiple, cumulative effects on consumer health and well-being, increasing the risk of both physical and psychiatric morbidity [9,10]. People with mental illness experience significantly higher rates of cardio-metabolic conditions [11], tobacco use, and obesity [12]. Fewer opportunities to engage in meaningful activities may also have a negative impact on valued life roles, social participation, quality of life, and full engagement in personal recovery [9,13,14,15]. 

### Interventions for Activity Participation in Recovery 

Given the serious consequences of decreased participation, several interventions have been developed to enable consumers to engage in meaningful activities. To date, these interventions have mostly focused on vocational participation. A survey of Australian mental health consumers found that 76% were unemployed [16], with both symptoms and stigma presenting challenges to obtaining or returning to vocational activities [17,18]. A wide range of interventions has been developed to promote employment participation for consumers, with the Individual Placement and Support model having the best-developed evidence base [19,20]. However, these interventions often presume consumers have a certain level of function and motivation as a prerequisite. Hitch, Vernon, and Dun [16] advocated for a scaffolded, recovery-aligned approach to activity participation for consumers, with targeted support at every point on the continuum from beginning daily activity engagement to workforce participation. 

Interventions that address the earlier phases of activity participation during recovery have begun to emerge in the past decade. The Action Over Inertia (AOI) [21] program was developed by Canadian occupational therapists to promote health and well-being by increasing the time spent by consumers on meaningful activity. The program [21] is provided over 8, 60 min sessions and can be delivered individually or to a group. AOI activities are supported by workbook resources, which provide a basis for collaborative work between the consumer and clinician. A multi-centered randomized controlled trial [22] found the program had clinical utility and was well received by both consumers and clinicians. A qualitative study of consumer and clinician experiences of AOI [23] also found that the program is flexible and supportive of consumers developing a greater understanding of their personal barriers and facilitators to activity participation. 

The Balancing Everyday Life (BEL) intervention [24] is a 12-session (plus 2 booster sessions) intervention that aims to enable consumers to achieve balance in their activity participation to support their recovery. These sessions include an educational component, group activity, and homework tasks where participants trial strategies to change their participation. Individualized goal setting and peer support are both encouraged throughout the program, and each session is co-facilitated by two health professionals. A cluster-randomized controlled study [24] found that consumers in the BEL program were able to achieve higher levels of activity engagement, balance, and well-being more quickly than those provided with treatment as usual.

The preliminary evidence for both AOI and BEL provides promising data regarding their impact on consumer participation. However, none of the available interventions address a complete continuum of activity participation from initial engagement through to workforce participation as part of personal recovery. Participants in these programs may therefore need access to additional interventions to meet their vocational needs, which may not provide a coherent or well-articulated pathway to their ultimate recovery goals. The aim of this study was to describe the outcomes of a group program designed to address all stages of activity participation during recovery, known as Pathways to Participation (P2P).

## 2. Materials and Methods

The study utilized a descriptive pilot longitudinal design to investigate outcomes for consumers participating in a single community mental health service. The study received ethical approval to proceed from the relevant Human Research Ethics Committee (HREC 2017.074).

### 2.1. Development of the P2P Program

The P2P intervention is a hybridized program that aims to enable consumers to engage in meaningful activities at all stages of recovery. The P2P program combines two evidence-based interventions, the Action Over Inertia (AOI) program and The WORKS program. In addition to the overview of AOI presented above, the program emphasizes gradual momentum towards personal recovery goals and aims to increase engagement, enhance activity balance, promote social interaction and community engagement, and improve overall health and well-being. 

The WORKS framework [25] was developed through active collaboration between consumers and clinicians in an English mental health service. It focuses on vocational participation and is accompanied by workbook resources that may be delivered either individually or to a group over six, two-hour sessions [26]. The WORKS was specifically designed for consumers who cannot access supported employment or are yet to decide whether they wish to work and has three potential starting points: Starting Out, Moving Forward, and Keeping Going and Growing [25]. The WORKS ‘Starting Out’ phase was chosen for incorporation into the P2P program because the initial stages of vocational participation were thought to best align with the content of AOI. Robertson [27] found that the Starting Out portion of the WORKS had a positive impact on consumer participation and was considered an acceptable intervention by participants. A further evaluation also found that the program provided a supportive environment for consumers considering and exploring vocational participation and that co-facilitation by occupational therapists and peer workers was particularly valued by all stakeholders [28,29].

The developers of both the AOI and WORKS programs encourage modifications for local conditions, and the first author undertook adaptations based on her previous experience of delivering these interventions in community mental health services. The model of co-facilitation by occupational therapists and peer support workers was retained from previous iterations of the WORKS, given the positive feedback provided by stakeholders of that program [30]. Consultation with local occupational therapists identified that 10 weeks would be the longest feasible program length, given available resources and the overall structure of the group program. The workbooks of both interventions were analyzed for similar and divergent content and their alignment to a scaffolded approach to activity participation and recovery (progressing from simple to more complex activities). A draft P2P program manual was developed and circulated to senior occupational therapists at the mental health service for consultation. Additional instructions and supporting content were added for some activities from the feedback received, but no changes to the structure of the program were considered necessary. 

Prior to the commencement of the P2P program at each site, facilitating occupational therapists and peer support workers were provided with intervention training based on the manual content. This training was instituted based on previous findings [30], which indicated that both occupational therapists and peer support workers needed more formalized and structured education before stepping into the role of group co-facilitator. After each session of P2P, facilitators undertook a debriefing and group process evaluation conversation with each other and/or the local chief occupational therapist. These debriefings were designed to provide an opportunity to reflect on how the group went, the group dynamics, discuss proposed adjustments to meet the needs of individuals and the group, and reflect on program fidelity. A mixed-methods implementation study was undertaken in parallel to the study reported here and is reported separately. 

### 2.2. Procedure 

A single iteration of the P2P program was delivered at each of the four catchment areas of the adult mental health service. Each co-facilitated session lasted for two hours, with the first four weeks based on AOI content and the final six weeks based on The WORKS content. An information session was provided to consumers prior to P2P commencement, ensuring they understood the commitment required before agreeing to attend. Consumers were encouraged to attend the P2P program every week; however, it was not compulsory, and anyone missing a session was provided with extra contact from the facilitators to support them to rejoin and catch up on content. An overview of the content of each session is provided below in Table 1, and further details of the content of each session is provided in the Appendix A. 

### 2.3. Data Collection

A total of five key outcomes were identified for investigation in this study: consumer identified needs, time use, self-rated recovery, psychosocial health, and community participation. Consumer identified needs were measured using the Camberwell Assessment of Need Short Appraisal (CANSAS) [31], which was designed for both research and clinical use with consumers. The tool assesses problems encountered over the past month in 22 domains of daily activities, with each domain scored on a scale of 0–2 (0 = no need, 1 = met need, 2 = unmet need). The CANSAS-P (patient version) was utilized in this study, which a previous Australian study [32] found to have good test reliability (particularly for unmet need) and acceptability. While a factor analysis determined that a four-factor structured model (social and cognitive functioning, emotional responsivity and coping with daily challenges) appears to fit this measure, it is most usually interpreted on an item by item basis [33]. 

Time use was measured via a 24 h time diary, in which consumers listed the activities they participated in, where it was performed and whom it was performed with at hourly intervals for the previous day. This method of gathering time use data is practiced across a range of disciplines, although it can be susceptible to recall bias [34]. Data regarding self-rated recovery was obtained using the Recovery Assessment Scale—Domains and Stages (RAS-DS), which is a self-reported 38-item questionnaire [35]. The RAS-DS includes four domains (personal, social, functional, and clinical recovery), with both total and domain scores calculated. The feasibility of this measure has been proven, and it is also reported to have excellent internal reliability and validity and good sensitivity to change [36]. 

Psychosocial health was measured using the Behavior and Symptom Identification Scale (BASIS-24) [37], which is designed to measure major symptoms and functional difficulties for consumers over the past week. The 24 items are rated on a Likert scale of 0 (no difficulty) to 4 (extreme difficulty), and the tool includes depression/functioning, interpersonal problems, self-harm, emotional lability, psychosis, and substance use subscales. Research has confirmed its utility in outpatient and residential settings, with psychometric testing showing it to have adequate validity, reliability, and sensitivity [38]. Community participation was investigated using the Living in the Community Questionnaire (LCQ) [39]. The LCQ is a measure of social participation, which collects data around self-rated participation in social activities, education, voluntary work, caring for others, employment, living situation, health, advocacy, outcomes, and recovery. The LCQ is reported to have sound psychometric properties, established with Australian samples [40]. 

In combination, the outcome measures used in this study took approximately 15 min to complete. Outcomes were measured at the beginning of the P2P program (T1), immediately post P2P program (T2), and 3 months post P2P program (T3). Demographic data were also collected for all consumers at recruitment; at subsequent time points, a further question regarding any other vocational interventions was included. Participating consumers were reimbursed for their time completing data collection tasks with shopping vouchers. 

### 2.4. Sample

Initially, presentations about the P2P program were provided to multidisciplinary teams at each local service. Posters and brochures were also distributed to multidisciplinary team members to provide to potential participants and were displayed in the waiting room of each service to enable consumer self-referral. Multidisciplinary team members were not expected to recruit consumers, only to provide them with information and support the consumer to complete a brief registration form if required. Participation in the P2P program was not contingent on participating in this study. 

All local community services within the catchment of the mental health service were included in this study, comprising Continuing Care Units, Prevention and Recovery Care Services, and Community Mental Health Teams. For inclusion in the study, consumers needed to be aged between 18 and 65 years old and be receiving care from one of the participating services at the time of recruitment. Other inclusion criteria were (1) generally stable health (i.e., no acute physiological or psychiatric illnesses requiring admission to acute services at the time of recruitment); (2) formal diagnosis with a serious mental illness or mental disorder with associated significant levels of disturbance and psychosocial disability; (3) spoken English to a level sufficient to participate in the group program without an interpreter; (4) capacity to independently provide informed consent, and (5) considered to present low to moderate risk by their key clinician. As co-morbidity was so prevalent amongst consumer populations [41], those who met all other inclusion criteria and had other diagnoses were included. For their data to be included in the analysis, consumers must have attended at least 7 of the 10 scheduled sessions. Consumers who had already undertaken the AOI or WORKS programs were excluded from recruitment. Any consumer experiencing a relapse during the P2P program was also excluded; however, all data collected to that point remained available for analysis. 

Once registered for the P2P program, the consumer was provided with information regarding this study from the first author at an information session for all group attendees prior to the program commencing. Individual, specific written consent was obtained for every consumer participant in this study. Consumers were provided with a plain language statement prior to consent being sought and were encouraged to ask questions from their supporters prior to confirming their intention to participate. Participating consumers could choose extended P2P session attendance, a separate clinic appointment, or a telephone interview to complete outcome measures at each time point. 

### 2.5. Data Analysis

Descriptive analysis was undertaken for all outcome measures collected in this study. Interval variables were analyzed using means, standard deviations, and ranges, while categorical variables were analyzed using frequencies and proportions. Where descriptive analysis indicated a large change in outcomes, analytical analysis was deployed using the Chi-square test. 

## 3. Results

### 3.1. Sample Characteristics 

A total of 17 consumers were recruited for this study; however, 6 were lost to follow-up prior to the end of the P2P program. The following analysis focuses on the 11 consumers for whom repeat measures were available at T2 and the 8 consumers who provided data at T3. Reasons for attrition from this study included gaining employment (*n* = 1), moving away (*n* = 2), and choosing not to continue (*n* = 6). Most consumers were female (*n* = 7, 63.64%). Mood disorders, anxiety disorders, and schizophrenia were the most prevalent psychiatric diagnoses, which consumers had been living with for 11.50 years (±7.74) on average. They also experienced other health conditions (including orthopedic, sensory, neurological, endocrine, and hematological diagnoses), which were also long-standing (M = 13.40, ±10.50). Around half of the sample (*n* = 5, 45.45%) had co-morbidities in addition to their mental illness. 

Most consumers (*n* = 8, 72.73%) were singled or divorced, gained income from welfare, had completed high school education, and did not have children. A similar proportion (*n* = 7, 63.64%) were living with family members and undertaking additional vocational interventions to the P2P program during the study. 

### 3.2. Consumer Identified Needs

Most consumers (*n* = 8, 72.73%) experienced a decrease in unmet needs over the course of this study. There was a significant change in the proportion of identified needs between T1 and T2, with met needs increasing and unmet needs decreasing, χ^2^ (1, N = 242) = 4.55, *p* = 0.033. Further decreases in both met and unmet needs were reported at T3, but these were not statistically significant. Information about conditions or treatment (*n* = 7, 63.64%) was the most commonly unmet need at T1, along with self-care (*n* = 5, 45.45%). Money was frequently identified at T1 (*n* = 6, 4.55%) and T2 (*n* = 8, 72.73%), while physical health (*n* = 9, 81.82%) and daytime activities (*n* = 8, 72.73%) were highlighted T2 (*n* = 5, 62.50%) and T3 (*n* = 7, 87.5%). Distress (*n* = 6, 75.00%) and psychotic symptoms (*n* = 5, 62.50) were only prevalent at T3. 

### 3.3. Time Use

While time used for sleep remained stable for most, some consumers increased or maintained participation in activities of daily living (*n* = 3), productivity activities (*n* = 2), and leisure activities (*n* = 4) during this study. Both positive and negative changes in activity participation times were recorded across the study, and there was considerable variability between individual consumers. The amount of time consumers spent in the community remained steady from T1 (M = 2.78, ±2.18), to T2 (M = 3.30, ±2.75), and T3 (M = 2.86, SD ± 2.85). As shown below in Table 2, leisure comprised the majority of participation in daily activities, with relatively few hours recorded for productive or social activities. 

### 3.4. Self Rated Recovery 

As shown in Figure 1, both mean total and social RAS-DS scores decreased between T1 and T2 before increasing again at T3. The sample mean total and personal recovery scores increased by 4% over the course of the study, while higher increases were identified for clinical recovery (6%) and social recovery (8%). 

A total of 7 consumers (63.64%) experienced an increase in total recovery scores between T1 and T2, while 5 consumers (62.5%) experienced further increase or maintenance of their total recovery scores between T2 and T3.

### 3.5. Psychosocial Health 

There were no statistically significant changes in total or subscale BASIS-24 scores over time for any participating consumers. Psychosocial health remained constant for consumers throughout the study, with any changes observed being small fluctuations in scores.

### 3.6. Community Participation and Well-Being 

The proportion of consumers rating their social participation as ‘about right’ increased by 46% between T1 and T2, with smaller increases reported for caring participation (17%) and work participation (17%). A further 25% of consumers rated their participation in unpaid work as ‘about right’ at T3; however, (as shown below in Figure 2) perceptions of participation in all other activities categories remained steady.

The proportion of consumers reporting good to excellent well-being also increased during the study, however not to a statistically significant degree. There was a steady increase in the proportion of consumers expressing a positive perception of their physical health (36%), having a say in the community (30%), and hopefulness (21%). However, fewer consumers felt positive about having a say within families, having opinions respected, happiness, goal achievement, and belonging after 3 months than at T2. 

## 4. Discussion

This study has described the outcomes of consumers participating in a group program designed to address all stages of activity participation during recovery, known as Pathways to Participation (P2P). While this pilot study can only provide a limited perspective on effectiveness, the findings presented here indicate the program did achieve some positive outcomes for the consumers who took part. 

P2P appeared to have the most positive impact on decreasing unmet needs for consumers and on social recovery and participation. The specific mechanisms by which consumers were able to meet their needs more effectively were not part of the data collection for this study but should be qualitatively investigated in future research. However, an Israeli study of the needs of people with serious mental illness [42] asserts that mental health services must engage with consumers’ most prominent needs to successfully promote recovery. Similarly, a recent scoping review has identified that recovery is enhanced by enabling social participation and interventions which include elements of peer or lived experience [43]. These findings, therefore, provide preliminary evidence that the P2P program successfully addresses aspects of recovery that matter to consumers and potentially increases the potential for positive outcomes. 

However, some of the improvements reported by consumers between baseline and the end of the program were not sustained over the following three months. The evaluation of the long-term effects of occupational interventions is important to understand whether they have had a meaningful impact on the consumer, particularly as a fading of reported effectiveness over time is reported in many therapeutic studies [44]. Given their central role in coordinating services and support, it is likely that key workers/case managers have a significant role in supporting consumers to sustaining activity and participation gains made during the P2P program. Most case managers or key workers are not occupational therapists and, therefore, may not have the same focus on activity and participation as a core part of their role. The inclusion of regular, structured follow-up post-P2P group, with both consumers and their key workers, is suggested for future iterations to prevent fading of positive outcomes over time. Beneficial peer supports formed during the P2P program could also be sustained by consumers continuing to meet as an informal support group, but this would need to be a matter of individual choice. 

This aspect of the finding also raises questions about whether a program like P2P is most appropriately delivered by occupational therapists employed by state-based statutory mental health services. The complexity of factors that influence the activity and participation of consumers indicates that longer-term psychosocial rehabilitation is needed to enable recovery. In recent years, rehabilitation has had less of a presence in Australian mental health policy, with much of the public discourse focused on the tension between resourcing acute and community-based services [45]. However, the number of people requiring state-based mental health services continues to rise [46], and reports persist of access barriers for people with mental illness to other forms of support, such as the National Disability Insurance Scheme [47]. Despite the challenges of providing this psychosocial rehabilitation program in an increasingly acuity-focused public mental health system, the potential benefits indicated in this study suggest it should be available for consumers engaging with this (and indeed other) service settings. 

While some positive outcomes were short-lived, no significant changes in other outcomes were identified during this study. Both time use and psychosocial health outcomes remained steady across all time points, which suggests the P2P program made no impact on either of them. However, the proportion of consumers rating their social participation and unpaid work as ‘about right’ increased, as did the number of participants identifying their well-being as good to excellent. The AOI program is a time-use intervention [22], and so these findings may be reflective of the small sample size. However, these findings can also be interpreted as demonstrating that consumers were able to maintain their health and well-being while participating in the P2P program. While some see maintenance as antithetical to recovery [48], it can also be a legitimate occupational choice for people with severe and persistent mental illness [49]. Findings from the implementation study arising from this pilot of the P2P program [50] suggest it exceeded the expectations of most participating consumers. A qualitative exploration of what constitutes a ‘good’ outcome from the P2P program from the consumer perspective is recommended for future research into this intervention. 

The findings of this study also highlight that recovery is a very individualized and personal process for consumers. Recovery is not a linear process, which is why recovery scales do not have ‘clinically significant’ change thresholds [36]. The trajectories of individual consumers, from baseline to the end of the program and beyond, described a range of recovery pathways. These outcomes from the P2P program could be more successfully captured using other research methods, particularly those adopting a mixed-method or case study approach. The prioritization of individual goal setting in this program also offers some challenges to program evaluation, as an important outcome for one consumer may not be important to another. Goal attainment scaling may be suitable for future evaluations of this and other similar programs, as may a multi-attribute utility instrument that focuses on activities and participation.

### Limitations

This study has several limitations that must be considered when interpreting the findings. The small sample size as a pilot study means that effectiveness could not be comprehensively assessed, and the participating consumers were not representative of any larger population. It was also completed within a limited geographical area and did not reflect prevailing changes to practice required since the COVID-19 pandemic. Aside from the methodology, there was also significant attrition from the study and loss of follow-up over time. This issue may be particularly problematic for mental health studies, as participants are experiencing a range of issues that may interfere with their ability to participate [51]. Despite these limitations, the study has provided evidence that the P2P program is feasible and identified some important considerations for future iterations. 

## 5. Conclusions

In conclusion, this study has described the outcomes experienced by consumers participating in pilot iterations of the Pathways to Participation (P2P) program. Some positive outcomes were identified regarding unmet needs and social participation, but no significant improvements were recorded in time use or psychosocial health. The findings presented here provide a basis for ongoing research into the effectiveness of the P2P program, which should include qualitative and mixed methodologies to better capture consumer experiences of the program.

## Figures and Tables

**Figure 1 ijerph-19-06088-f001:**
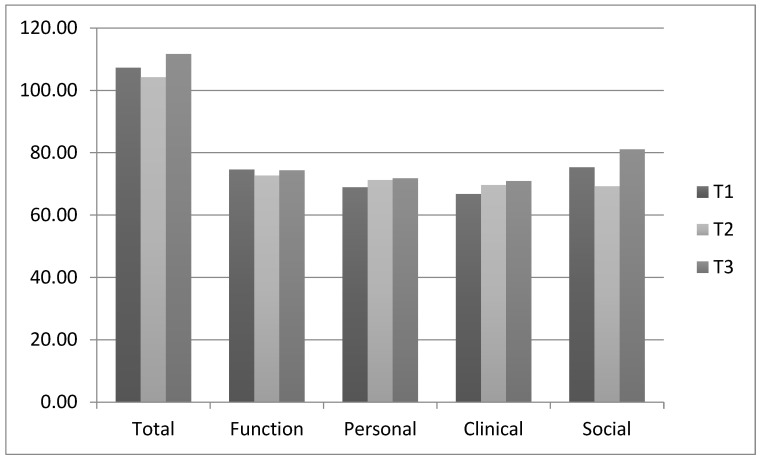
Sample Mean RAS-DS scores.

**Figure 2 ijerph-19-06088-f002:**
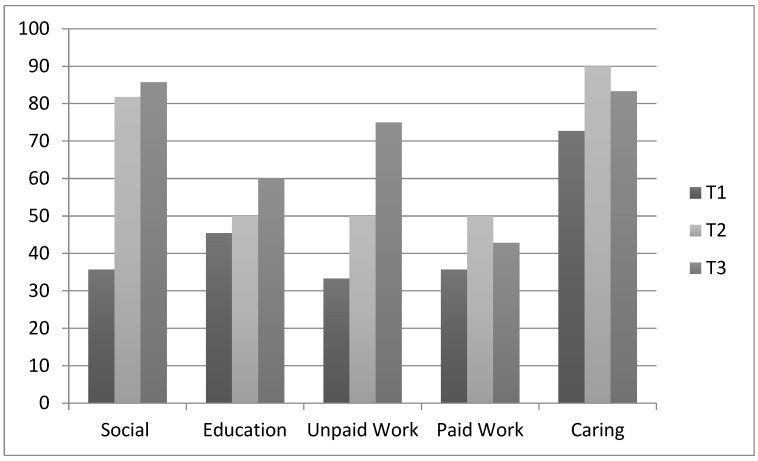
Proportion of consumers perceiving their participation as ‘About Right’.

**Table 1 ijerph-19-06088-t001:** Overview of Pathways to Participation (P2P) program content.

Week	Content
1	Introduction to the program Completion of self-profile Homework: Daily time use diary
2	Typical day and current balance of activity Finding meaning and satisfaction in activity Social interaction and accessing the community through activity Activity engagement measure Activity patterns I would like to change
3	The benefits of my current activities My personal qualities and traits What skills do I have? What does work/education/volunteering mean to me?
4	Reducing stress in activity participation The health and well-being benefits from my current activity patterns
5	Preparing for change in activity participation and prioritising Planning for activity change Homework: Thinking about your week sheet
6	What stops me from moving on, and how to overcome these barriers Homework: Record of activity experiments
7	Identifying and reflecting on changes in my activity patterns Homework: My energy levels sheet
8	Dealing with fatigue and maintaining a work/life balance Homework: Daily planner
9	Revisiting my self-profile Things to write on my resume Homework: Drafting a resume
10	Moving on to your next phase Guest speaker from a local employment agency What did you like or enjoy? Presentation of certificate of completion

Treatment as usual for each consumer continued alongside his or her participation in the P2P program, including access to services designed to increase their activity and vocational participation and promote recovery.

**Table 2 ijerph-19-06088-t002:** Consumer Time Use in Average Hours Per Day.

Activity Category	Time Use (Hours Per Day) Mean and Standard Deviation		
	T1	T2	T3
Sleep	10.56 (±4.13)	10.30 (±1.70)	10.25 (±1.85)
Personal ADL	2.72 (±1.56)	2.40 (±0.97)	2.89 (±0.91)
Instrumental ADL	2.94 (±2.72)	3.10 (±2.77)	2.71 (±1.70)
Education/Work	0.83 (±1.27)	1.30 (±2.36)	0.86 (±2.27)
Leisure	6.56 (±2.89)	6.80 (±3.33)	6.14 (±3.13)
Social	0.72 (±0.97)	0.10 (±0.32)	1.14 (±2.19)

Note: ADL = activities of daily living.

## Data Availability

The data presented in this study are available on request from the corresponding author. The data are not publicly available due to participants not having given their consent for this form of disclosure.

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
