# Peer review of "The Pathways to Participation (P2P) Program: A Pilot Outcomes Study"

_ijerph, 2022, doi:10.3390/ijerph19106088_

Round 1
Reviewer 1 Report
This study explored the outcomes of participation in a group program (P2P) among people with mental illness. The outcomes including consumer identified needs, time use, self-rated recovery, psychosocial health, and community participation were explored during and after the participation in P2P program. Through the idea of study is interesting, the result contains a serious limitation. This is a few number of participants, only 17 participants, and 8 participants left at the end of the program. I have following comments.
- Due to a very few number of participants, I think authors need to include more secondary sources to support the results, and make clearer scientific discussions. Otherwise, the results are not convincing, and the contribution of this research is limited.
- please clearly explain characteristics of P2P program such as characteristics of activities in each session. In each session, what participants were asked to do and how long they were asked to do. This is important. In case that other relevant studies would like to employ this program.
- Please indicate date (month and year) which data were collected.
- Please also discuss why those 5 outcomes were selected to explore, and how they are likely to be achieved after the participation in P2P. These discussions would lead to the development of theoretical assumptions in this study.
- Regarding the result of exploration on psychosocial health, thought, the result was not statistically significant, the result must be presented as well. Actual scores of psychosocial health, and the result of chi-square test should be reported.
- The result of statistic test for communication participation must be reported. Though, they were not statistically significant.
- Regarding self-related recovery, please discuss why mean total, function, and social RAS-DS scores decreased between T1 and T2. In addition, if the statistical difference was tested, please also report the result.
- Surprisingly, many assumptions were not justified by the empirical survey. These results must be discussed in details. For instance, time use, psychosocial health, and community participation were not significantly changed, and the results of statistic tests were not significant. These results must be explained based on theoretical perspectives. Finally, it would be more appropriate to provide implications for the improvement of P2P.
- There was no a significant change in time use. Please explain characteristics of proper time use and improper time use, and discuss whether time use reported by participants was good or not, and how P2P program could help improving.
Reviewer 2 Report
Thank you for the submission of the manuscript. The manuscript is informative and interesting.
I found the aim of the paper to be clearly identified. The introduction provides the reader with essential information for understanding the study.
The writing style found in the manuscript is appropriate for clarity and meaning. I found the citations to be appropriate. However, approximately 63% (approximately 32 of 51) of the references are outside of the five-year window: but, I believe the references cited are necessary to explain and support the methods and methodology.
Reviewer 3 Report
Dear author/s, I have read your article with great interest. The topic is very important and up-to-date. The manuscript is well positioned, with a clear aim and contributions. You analyzed the relevant literature and presented the methodology and research results in a very clear way. Then you discussed the results and compared them with the results of other studies. Additionally, a longitudinal study is a great added value. The manuscript is written in correct English. Taking all this into account, I recommend this manuscript for publishing in its current form.
Author Response
No response required as no additional recommendations made
Reviewer 4 Report
Estimated Authors,
I've read your paper with great interest. In this study, reporting on outcomes of participation in a group program designed to address all stages of activity participation (i.e. Pathways to Participation, P2P). Authors were able to identify a substantial reduction in unmet needs and improvements in self-rated recovery scores. However, these results were somewhat impaired by the lack of changes in either time use or psychosocial health.
In fact, these results are quite interesting because of the innovative intervention the Authors reported on. From the point of view of this reviewer, the main shortcoming of this study is represented by the reduced number of participants, with subsequent limits in the generalizability of the results. However, such issues are properly assessed and discussed.
In summary, I've no specific requests, and I'm recommending the eventual acceptance of the study.
Author Response
Thank you for your kind words. As you note, we have acknowledged the limitations posed by the small number of participants but unfortunately that cannot be helped. We have learnt a lot from this study which will enable us to improve our recruitment methods and approaches in future.